# In Vitro Assay Using *Proboscidea parviflora* W. and *Phaseolus lunatus* L. Plant Extracts to Control *Pythium amazonianum*

**DOI:** 10.3390/microorganisms12061045

**Published:** 2024-05-22

**Authors:** Yisa María Ochoa Fuentes, Antonio Orozco Plancarte, Ernesto Cerna Chávez, Rocío de Jesús Díaz Aguilar

**Affiliations:** 1Departamento de Parasitología Agrícola, Universidad Autónoma Agraria Antonio Narro, Calzada Antonio Narro 1923, Col. Buenavista, Saltillo 25315, Coahuila, Mexico; yisa8a@yahoo.com; 2CULTA S. A. de C. V. Blvd. Luis Echeverria Álvarez No. 1700, Col. Altavista, CD. Mante 89880, Tamaulipas, Mexico; antonioorozco25@outlook.es

**Keywords:** plants, extracts, avocado, *Phytophythium* sp.

## Abstract

Avocado tree wilt is a disease caused by *Phytophthora cinnamomi* Rands. Recently, this disease has been associated to *Pythium amazonianum*, another causal agent. Avocado tree wilt is being currently controlled with synthetic fungicides that kill beneficial microorganisms, polluting the environment and leading to resistance problems in plant pathogens. The current research work aims to provide alternative management using extracts from *Proboscidea parviflora* W. and *Phaseolus lunatus* L. to control the development of mycelia in *P. amazonianum* in vitro. Raw extracts were prepared at UAAAN Toxicology Laboratory, determining the inhibition percentages, inhibition concentrations and inhibition lethal times. Several concentrations of the plant extracts were evaluated using the poisoned medium methodology, showing that both extracts control and inhibit mycelial development, in particular *P. lutatus*, which inhibits mycelial growth at concentrations lower than 80 mg/L, being lower than *P. parviflora* extracts. These extracts are promising candidates for excellent control of *Pythium amazonianum*.

## 1. Introduction

Mexico is a global producer of avocado (*Persea americana* Mill.) and a consumer leader. There are 16,436,760 ha currently planted with avocado in the country, yielding an annual production of 10,686,454 tons. The states with higher levels of production are Michoacan, Jalisco, Estado de Mexico and Nayarit [1]. Among the most important diseases, avocado tree wilt caused by *Phythopthora cinnamomi* Rands is the main disease affecting this crop worldwide [2], besides having more than one thousand host plant species [3]. In 2016, *Pythium amazonianum* = *Phytophytium* sp. *amazonianum* [4,5,6] was reported for the first time as the causal agent of avocado wilt in Peribán Michoacán, Mexico. Molecular and morphological characterization describes *Phytophytium mirpurense* as a phylogenetically separate entity from *Phythium* and *Phythopthora*, since it is the only entity with internally proliferating papillate oogonia and cylindrical or lobed antheridia associated with the disease. In order to face the problems caused by this disease, avocado growers have been searching for management alternatives. However, synthetic fungicides provide poor control over this disease and the use of multiple applications leads to different problems like human toxicity [7,8,9], export rejects due to chemical residues, environmental damage [10] and impact on beneficial fauna [11]. Likewise, plant pathogens can develop resistance to active ingredients in synthetic fungicides, leading to efficiency problems and [12] the use of higher rates, or the development of new agricultural chemicals that are intended to replace those products to which fungi have shown resistance [13]. Due to the problems caused by the indiscriminate use of synthetic fungicides, it is necessary to develop natural alternatives to control fungal diseases. One of those strategies is the use of plant extracts, which are organic, biodegradable and harmless to human health and the environment [14]. However, the use of extracts is scarce due to the limited availability of products in the market, the difficulties in their formulation and the decrease in their efficacy in the field [15]. The use of plant extracts is an alternative for integrated crop management. Due to their low cost and their potential, plant extracts can be used to control and inhibit bacteria, as well as plant pathogenic fungi [16].

Furthermore, it is mentioned that Proboscidea parviflora W ethanolic and methanolic extracts have fungicidal effects that inhibit the growth of Aspergillus niger, Aspergillus flavus, Penicillium chrysogenum, Penicillium expansus, Fusarium poae and Fusarium moniliforme; although we do not know which metabolites are present in these species [17].

*Proboscidea lousiana* contains 47% triterpenes and 38% glucosiloxy-fatty acids [18]; phytoalexins have been found in *Phaseolus vulgaris* beans [19]. Phytoalexins have low molecular weight and are secondary metabolites of diverse chemical nature and antimicrobial activity, including terpenoids, alkaloids, glycosides and flavonoids, among others. These metabolites offer protection against pathogens and they can contribute directly to control of bacterial and plant pathogenic fungal growth. This research work seeks management alternatives, using highly effective inputs with lower environmental impact that can be part of *Pythium amazonianum* management practices. The aim was to prepare and assess the in vitro effect of *Proboscidea parviflora* W. and *Phaseolus lunatus* L. plant extracts on pathogens that affect avocado.

## 2. Materials and Methods

This research work was conducted at the Agricultural Parasitology Department of Universidad Autónoma Agraria Antonio Narro (UAAAN), Saltillo, Coah, Mexico.

### 2.1. Biological Material

The strain of *Pythium amazonianum* used in this experiment was isolated, purified and identified at UAAAN Toxicology Laboratory by [4]. The oomycete strains were white with aerial, cottony mycelium; microscopically the mycelium was hyaline, coenocytic, the oogonia with antheridia and spherical and globular sporangia.

### 2.2. Preparation of Assessed Extracts

*Proboscidea parviflora* W. was collected from Calera and Villa de Cos municipalities at Zacatecas, Mexico. The seed of *Phaseolus lunatus* L came from the growing region of San Miguel Totolapan municipality in Guerrero; these seeds were planted and cultivated under greenhouse conditions at UAAAN from October 2015 to January 2016. The extracts were prepared using different parts of the plant like roots, stems, leaves, flowers, green pods and seeds. The solvent was sterile distilled water, hexane and ethanol at 96%. A total of 200 g macerated plant material with 200 mL of sterile distilled water was mixed in a conventional blender (Oster). The samples were suspended in 800 mL of the solvent, obtaining a final volume of 1000 mL. All the preparations were stored in darkness at room temperature and they were stirred once a day for 30 days. After that time, the samples were filtered to separate the plant residues. The extracts were concentrated in a rotary evaporator (BUCHI R-200 Heathing Bath B-490, Marshall Scientific, Hampton, NH, USA) using the following temperatures: aqueous extracts at 90 °C, ethanolic extracts at 70 °C and hexanolic extracts at 60 °C. The samples of the aqueous macerated extracts were filtered. Raw extracts were kept in 30 mL amber-colored glass containers at 4 °C to preserve them.

### 2.3. Extracts’ ASSAY

Sixty-three extracts were evaluated using an in vitro bioassay technique (poisoned medium technique) with four replicates per dose. The raw extract was used to obtain the different concentrations. When PDA (BD Bioxon, Franklin Lakes, NJ, USA) reached a temperature between 35 °C and 40 °C, the extract quantities were added at the desired concentration. Explants with a diameter of 0.5 cm were introduced in sequence with the plant pathogens and they were incubated at 25 ± 2 °C in darkness (Lab-line Model 150, Barnstead, NH, USA) until the mycelia grew in the Petri dish that was used as a check test (PDA without extract), covering all the plate. Mycelial growth was measured every day with a digital vernier caliper and the readings were used to calculate the inhibition percentages, using the [20,21] formula:(1)$$\%\;inhibition=\frac{Mycellial\;growth\;of\;the\;check\;test-mycellial\;growth\;of\;the\;treatment}{mycellial\;growth\;of\;the\;check\;test}\times 100$$

### 2.4. Determination of Compounds in Extracts by Chromatography–Mass Spectrometry

The extracts with the greatest inhibitory effect were analyzed by chromatography and mass spectrometry in the Metabolomics and Mass Spectrometry Laboratory at LANGEBIO-CINVESTAV.

### 2.5. Analysis of the Results

The inhibition percentages obtained from the preliminary tests that were carried out to determine ${Ci}_{50}$, ${Ci}_{70\;}$ and ${Ci}_{90}$ were analyzed using R statistical system, version 3.3.2 by Probit regression. On the other hand, the highest concentrations per extract (mg/L) obtained from the mycelial growth inhibiting percentages were studied by Probit regression using statistical system R, Version 3.3.2, in order to determine the mean lethal time.

## 3. Results

Sixty-three extracts from different parts of the plant (Table 1) were prepared. Eight of them showed an inhibitory effect over *Pythium amazonianum* at different concentrations.

The preliminary test to determine the inhibitory concentrations (*Ci*) of the extracts included a biological window per extract (0.05, 0.1, 1, 10, 20, 40, 60, 80, 100, 400, 800, 1000, 2000, 6000 and 10,000 mg/L). Eight extracts showed inhibiting effects at the following rates: ethanolic of flower of *P. parviflora* (E.F.P.p), 2000, 2104.75 and 2265.76 mg/L; ethanolic of root of *P. parviflora* (E.R.P.p), 211.59, 691.6 and 1000 mg/L; aqueous filtered of green Pod of *P. parviflora* (F.A.V.v.P.p), 1, 2 and 4 mg/L; hexane of green pod of *P. parviflora* (H.V.v.P.p), 264.62, 725.67 and 2935 mg/L; aqueous filtered of leaves of *P. lunatus* (F.A.H.P.l), 0.05, 0.5 and 1 mg/L; aqueous filtered of seed of *P. lunatus* (F.A.S.P.l), 0.05, 0.1 and 1 mg/L; aqueous filtered of stem of *P. lunatus* (F.A.T.P.l), 10, 20 and 60 mg/L; and aqueous macerated of seed of *P. lunatus* (M.A.S.P.l), 0.1, 1 and 2 mg/L. Sixty-three extracts from different parts of the plant (Table 1) were prepared. Eight of them showed an inhibitory effect over *Pythium amazonianum* at different concentrations.

*P. lunatus* extracts F.A.H.P.l, F.A.S.P.l and F.A.T.P.l y M.A.S.P.l (Table 2) showed better inhibition rates. At concentrations of 0.05 mg/L (F.A.H.P.l y F.A.S.P.l), they inhibited 11.9 and 11.26% of *P. amazonianum* mycellian growth and, at 2 mg/L (M.A.S.P.l), they inhibited 100%, except in the case of F.A.T.P.l, where concentrations greater than 60 mg/L were required to inhibit 93.3% of the pathogen’s growth. None of these extracts exceeded 100 mg/L in concentration. *P. parviflora* extracts (F.A.V.v.P.p) had a behavior similar to the behavior of *P. lunatus* extracts, presenting an inhibiting effect at low concentrations (1, 2 and 4 mg/L), controlling mycelial growth at 27.4, 70.33 and 74.73%. E.R.P.p controlled 100% at 1000 mg/L, while E.F.P.p and H.V.v.P.p required concentrations over 1400 mg/mL to inhibit 100% of the pathogen’s mycelial growth. The inhibition percentage results differ from the data reported by [16]. Their research reported growth inhibition percentages (86.6%) of *Fusarium poae* under the ethanolic effects of *P. parviflora* prepared at 6% concentration. These results explain the low concentration requirements, without neglecting the fact that *F. poae* belongs to true deuteromycets fungi, with ketin cell walls (N-acetylglucosamine polymer), whereas the wall in oomycetes is made of β-1,3 y β-1,6 glycans [22].

*P. parviflora* extracts (E.F.P.p, E.R.P.p and H.V.v.P.p) showed a lethal time of 50% inhibition over the growth of *P. amazonianum* within the first 19.57 h, reaching 95% inhibition after 116.10 h. F.A.V.v.P.p inhibited 50% after 95.37 h and 95% after 258.86 h. *P. lunatus* extracts (F.A.T.P.l, F.A.H.P.l and F.A.S.P.l) showed a similar effect to F.A.V.v.P.p but required a longer time to inhibit 50% of the mycelial growth, opposite to M.A.S.P.l, which, just like E.F.P.p, E.R.P.p and H.V.v.P.p, required less time to control 50% of the mycelial growth. We observed that higher rates of the plant extracts inhibited more than 50% of the mycelial growth during the first 24 h, except M.A.S.P.l, which has the same effect at low concentrations (Table 3).

At ${Ci}_{50}$, ${Ci}_{70\;}$ and ${Ci}_{90}$ *P. parviflora* extracts showed the following behavior: E.F.P.p showed an inhibitory effect within the concentrations proposed by Probit assay (Table 4), coinciding with the inhibition percentages (Table 1). According to Probit regression curve displacement (Figure 1), there is no need to exceed 2200 mg/L to inhibit 90% of the pathogen’s mycelial growth. H.V.v.P.p behaved similarly, without exceeding those concentrations. However, according to Probit analysis, E.R.P.p and F.A.V.v.P.p (Table 4) required slightly higher rates to achieve similar inhibiting effects. On the other hand, *P. lunatus* extracts (F.A.H.P.l, F.A.S.P.l, F.A.T.P.l and M.A.S.P.l) are similar to *P. parviflora* extracts (E.R.P.p and F.A.V.v.P.p), where Probit analysis (Table 4) suggested using slightly higher concentrations without exceeding 100 mg/L. Regarding our inhibition concentration estimations, we agree with the proposal of [23], who mentioned that, using a by-product of the agricultural industry (wine vinasse) at concentrations of 50 and 70 mg/L, it is possible to inhibit 100% of *Pythium aphanidermatum* and *Phytophthora parasitica* growth. Likewise, [24] studied a methanolic extract of *Origanum mejorana* L. at 8000 mg/L and a synthetic product (Metalaxyl at 750 mg/L) to inhibit *Phytophthora infestans* at 100%.

The extracts with the highest inhibitory effect were F.A.S.P.l and H.V.v.v.P.p, which were analyzed by a positive and negative mode scanning system using chromatography–mass spectrometry (Table 5 and Table 6).

## 4. Conclusions

Only eight extracts showed significant inhibitory effect. *P. lunatus* showed out-standing control of the plant pathogen at concentrations lower than 80 mg/L, while higher concentrations of *P. parviflora* were required. In regards to mean lethal time, *P. parviflora* extracts required less time to produce inhibitory effects, while *P. lunatus* extracts required longer for producing antifungal response. These extracts are promising candidates for excellent control of *Pythium amazonianum*; however, it is suggested to continue evaluating these extracts for the control of various phytopathogenic fungi and oomycetes.

## Figures and Tables

**Figure 1 microorganisms-12-01045-f001:**
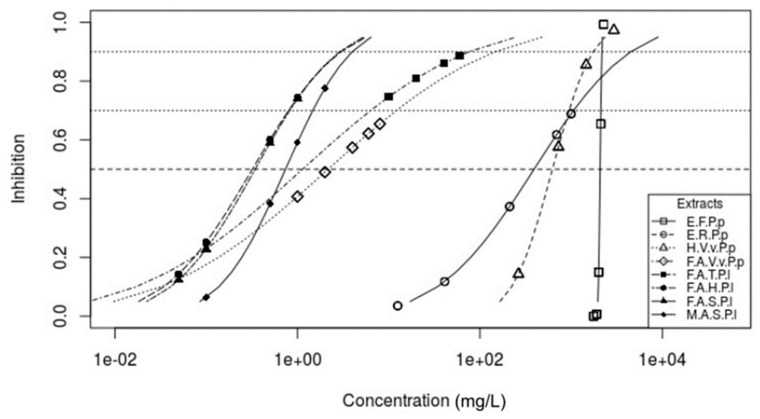
Probit regression depicting the inhibition of *Pythium amazonianum* mycelial growth under the effects of *Proboscidea parviflora* W. and *Phaseolus lunatus* L. Inhibition vs. Concentration.

**Table 1 microorganisms-12-01045-t001:** Preparation methods of the different extracts of *Proboscidea parviflora* W. and *Phaseolus lunatus* L.

Plant	Part of Plant	Solvent	Rest and Agitation Time	Temperature
***Proboscidea parviflora*** **W.**	Root, Stem, Leave, Flower and Green pods.	Water	30 days	90 °C
		Ethanol	30 days	70 °C
		Hexane	30 days	60 °C
		Aqueous filtered	30 days	0 °C
		Ethanolic Filtered	30 days	0 °C
		Hexanolico Filtered	30 days	0 °C
		Aqueous macerated	0 days	0 °C
***Phaseolus lunatus*** **L.**	Root, Stem, Leave and Seeds.	Water	30 days	90 °C
		Ethanol	30 days	70 °C
		Hexane	30 days	60 °C
		Aqueous filtered	30 days	0 °C
		Ethanolic Filtered	30 days	0 °C
		Hexanolico Filtered	30 days	0 °C
		Aqueous macerated	0 days	0 °C

**Table 2 microorganisms-12-01045-t002:** Inhibitory effect (%) over *Pythium amazonianum* mycelial growth of *Proboscidea parviflora* W. and *Phaseolus lunatus* L. plant extracts after 120 h.

Extracts	Concentration mg/L	Inhibition %	Standard Deviation
**E.F.P.p**	2000	2.14	1.60
	2104.75	62.09	11.27
	2265.76	100	0
**E.R.P.p**	211.59	17	7.38
	691.6	52.2	14.61
	1000	100	0
**F.A.V.v.P.p**	1	27.4	9.70
	2	70.33	3.53
	4	74.73	2.11
**H.V.v.P.p**	264.62	14.56	8.09
	725.67	57.93	15.27
	2935	100	0
**F.A.H.P.l**	0.05	11.9	2.91
	0.5	63.86	0.86
	1	78.33	2.35
**F.A.S.P.l**	0.05	11.26	2.21
	0.1	25.36	6.08
	1	75.03	0.96
**F.A.T.P.l**	10	80.7	5.00
	20	83.43	2.52
	60	93.3	0.38
**M.A.S.P.l**	0.1	13.7	4.66
	1	47.03	9.43
	2	100	0

E.F.P.p: ethanolic of flower of *P. parviflora*; E.R.P.p: ethanolic of root of *P. parviflora*; F.A.V.y.P.p: aqueous filtered of green pod of *P. parviflora*; H.V.v.P.p: hexane of green pod of *P. parviflora;* F.A.H.P.1: aqueous filtered of leaves of *P. lunatus*; F.A.S.P.1: aqueous filtered of seed of *P. lunatus*; F.A.T.P.1: aqueous filtered of stem of *P. lunatus*; M.A.S.P.1: aqueous macerated of seed of *P. lunatus*.

**Table 3 microorganisms-12-01045-t003:** Mean lethal time (TL50) of *Pythium amazonianum* inhibition under the influence of Pro*boscidea parviflora* W. and *Phaseolus lunatus* L. extracts.

*Extracts*	*Df*	*Tl50*	*Ltl*	*Utl*	*Tl05*	*Tl95*	*Intercept*	*Slope*	*p Value*
*E.F.P.p*	19	19.57	17.87	21.20	3.30	116.10	−2.74	2.12	2.02 × 10^−14^
*E.R.P.p*	19	19.57	17.87	21.20	3.30	116.10	−2.74	2.12	2.029 × 10^−14^
*F.A.V.v.P.p*	19	95.37	91.50	99.76	35.13	258.86	−7.50	3.79	7.31 × 10^−84^
*H.V.v.P.p*	19	19.57	17.87	21.20	3.30	116.10	−2.74	2.12	2.02 × 10^−14^
*F.A.H.P.l*	19	62.94	59.70	66.31	11.57	342.22	−4.02	2.23	7.66 × 10^−90^
*F.A.S.P.l*	19	72.63	69.89	75.53	15.79	333.88	−4.62	2.48	2.535 × 10^−15^
*F.A.T.P.l*	19	28.29	26.57	29.93	3.88	205.97	−2.76	1.90	8.50 × 10^−19^
*M.A.S.P.l*	19	19.57	17.87	21.20	3.30	116.10	−2.74	2.12	2.02 × 10^−14^

Df: degrees of freedom, TL: lethal time, LTL: lower time limit and UTL: upper time limit. E.F.P.p: ethanolic of flower of *P. parviflora*; E.R.P.p: ethanolic of root of *P. parviflora*; F.A.V.y.P.p: aqueous filtered of green pod of *P. parviflora*; H.V.v.P.p: hexane of green pod of *P. parviflora*; F.A.H.P.1: aqueous filtered of leaves of *P. lunatus*; F.A.S.P.1: aqueous filtered of seed of *P. lunatus*; F.A.T.P.1: aqueous filtered of stem of *P. lunatus*; M.A.S.P.1: aqueous macerated of seed of *P. lunatus*.

**Table 4 microorganisms-12-01045-t004:** Inhibitory concentrations (${Ci}_{50}$, ${Ci}_{70}$ and ${Ci}_{90}$) of *Proboscidea parviflora* W. and *Phaseolus lunatus* L. extracts over *Pythium amazonianum* mycelial growth.

EXTRACT	Ci	Concentration mg/L	Lower Fiducial Limits 95%	Upper Fiducial Limits 95%
**E.F.P.p**	${Ci}_{50}$	2075.19	2034.05	2117.33
	${Ci}_{70}$	2114.29	2076.48	2172.06
	${Ci}_{90}$	2172.05	2126.83	2266.83
**E.R.P.p**	${Ci}_{50}$	391.65	231.32	766.67
	${Ci}_{70}$	1062	575.68	3074
	${Ci}_{90}$	4483	1819	26939
**F.A.V.v.P.p**	${Ci}_{50}$	2.16	0.38	3.89
	${Ci}_{70}$	12.14	5.89	909.51
	${Ci}_{90}$	146.32	26.82	2716
**H.V.v.P.p**	${Ci}_{50}$	622.12	516.39	734.22
	${Ci}_{70}$	948.19	803.43	1142
	${Ci}_{90}$	1742	1411	2330
**F.A.H.P.l**	${Ci}_{50}$	0.32	0.28	0.36
	${Ci}_{70}$	0.79	0.68	0.95
	${Ci}_{90}$	2.98	2.32	4.04
**F.A.S.P.l**	${Ci}_{50}$	0.34	0.30	0.38
	${Ci}_{70}$	0.82	0.70	0.97
	${Ci}_{90}$	2.89	2.27	3.84
**F.A.T.P.l**	${Ci}_{50}$	1.13	0.21	2.56
	${Ci}_{70}$	6.33	2.93	9.46
	${Ci}_{90}$	75.75	52.53	150.75
**M.A.S.P.l**	${Ci}_{50}$	0.73	0.50	1.10
	${Ci}_{70}$	1.47	0.99	2.81
	${Ci}_{90}$	3.97	2.23	12.78

Ci (50, 70, 90)—inhibitory concentration in mg/L; E.F.P.p: ethanolic of flower of *P. parviflora*; E.R.P.p: ethanolic of root of *P. parviflora*; F.A.V.y.P.p: aqueous filtered of green pod of *P. parviflora*; H.V.v.P.p: hexane of green pod of *P. parviflora*; F.A.H.P.1: aqueous filtered of leaves of *P. lunatus*; F.A.S.P.1: aqueous filtered of seed of *P. lunatus*; F.A.T.P.1: aqueous filtered of stem of *P. lunatus*; M.A.S.P.1: aqueous macerated of seed of *P. lunatus*.

**Table 5 microorganisms-12-01045-t005:** Determination of compounds present in the aqueous filtered extract of *P. lunatus* L. (F.A.S.P.l) seed in positive mode and negative.

	POSITIVE MODE			NEGATIVE MODE		
	COMPOUND	CHEMICAL FORMULA	RETENTION TIME (min)	COMPOUND	CHEMICAL FORMULA	RETENTION TIME (min)
A	Cycasin	C_8_H_16_N_2_O_7_	1.21	1-[(5-Amino-5-carboxypentyl)amino]-1-deoxyfructose	C_12_H_24_N_2_O_7_	1.21
B	Asparaginyl-Glycine	C_6_H_11_N_3_O_4_	16.36	Glutamyl-Asparagine	C_9_H_14_N_3_O_6_-	0.83
C	Avocadene 2-acetate	C_19_H_36_O_4_	20.73	1,3-Octadiene	C_8_H_14_	0.87
D	Gabapentin	C_9_H_17_NO_2_	23.98	Estriol-3-glucuronide	C_24_H_32_O_9_	1.93
E	Pentadecanoylglycine	C_17_H_33_NO_3_	30.34	Nebularine	C_10_H_12_N_4_O_4_	2.00
F	Phosphoglycolic acid	C_2_H_5_O_6_P	0.75	Imidazolelactic acid	C_6_H_8_N_2_O_3_	3.09
G	Biphenyl	C_12_H_10_	26.86			
H	(S)-Isosclerone	C_10_H_10_O_3_	0.87			

**Table 6 microorganisms-12-01045-t006:** Chromatography and mass spectrometry positive and negative mode of the aqueous filtered extract of *P. lunatus* L. (F.A.S.P.l) seed.

	POSITIVE MODE			NEGATIVE MODE		
	COMPOUND	CHEMICAL FORMULA	RETENTION TIME (min)	COMPOUND	CHEMICAL FORMULA	RETENTION TIME (min)
A	Dimethindene	C_20_H_24_N_2_	11.30	Cyclocurcumin	C_21_H_20_O_6_	11.09
B	1-Hydroxyacorenone	C_15_H_22_O_3_	4.96	(S)-Reticuline	C_19_H_23_NO_4_	1.30
C	Geranylacetone	C_13_H_22_O	11.24	Austdiol	C_12_H_12_O_5_	10.98
D	Dehydroisochalciporone	C_16_H_19_NO	11.61	2-Isopropyl-1,4-hexadiene	C_9_H_16_	0.87
E	2-Methylphenyl 2-methylpropanoate	C_11_H_14_O_2_	11.76	9S,10S,11R-trihydroxy-12Z-octadecenoic acid	C_21_H_34_NO_3_^+^	16.86
F	Valyl-Hydroxyproline	C_10_H_18_N_2_O_4_	11.80	N-Methylschinifoline	C_16_H_17_NO_2_	20.70
G	Unknown	C_12_H_16_N_2_O_4_	11.80			
H	Unknown	C_10_H_17_O_5_-	12.14			

## Data Availability

The original contributions presented in the study are included in the article, further inquiries can be directed to the corresponding author/s.

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
