# Peer review of "In Vitro Assay Using Proboscidea parviflora W. and Phaseolus lunatus L. Plant Extracts to Control Pythium amazonianum"

_microorganisms, 2024, doi:10.3390/microorganisms12061045_

Round 1
Reviewer 1 Report (Previous Reviewer 2)
Comments and Suggestions for Authors
The manuscript investigated the extracts from Proboscidea parviflora and Phaseolus lunatus on mycelial development of Pythium amazonianum. Several concentrations of the plants extracts were evaluated using the poisoned medium methodology, showing that both extracts control and inhibit mycelial development, in particular P. lutatus, which inhibits mycelial growth at lower concentrations than P. parviflora extracts.
This is a resubmission. I suggested rejection previously, because that no chemical name or structure has been indentified from the extracts. Now in the revision, the compounds present in the aqueous filtered extract of P. lunatus seed in both positive mode and negative mode are shown.
And all other criticisms have been answered or corrected. So I think that the current version may be ready for publication.
Author Response
I send manuscript with corrections.
Ernesto Cerna

Reviewer 2 Report (New Reviewer)
Comments and Suggestions for Authors
In this study, authors aimed to provide a management options using extracts from Proboscidea parviflora and Phaseolus lunatus L. to control the development of Pythium amazonianum mycelia.
Introduction is weak, need more text to emphasise the lack of effective BCA against Phytium species.
The study experimental design and statistical analyses is quite simple but acceptable. Provide more description on chromatography and mass spectrometry.
The results of this study contain some valuable elements. Authors showed that P. lunatus showed a significant control of the plant pathogen at concentrations lower than 80 mg/L; while higher concentrations of P. parviflora were required. Mean lethal time was lower for P. parviflora extracts in order to provide inhibitory effects compared to P. lunatus.
The Discussion is not acceptable. Discussion part is very weak. I suggest to separate Results and Discussion and provide a more deep comparison of the results with the previous studies. I suggest to prepare a Conclusion section with bullet points and also authors should address the limitations of their study and briefly discuss future needs for this topic.
The number of references can be increased up to 40. Formatting of this section is inconsistent. Some inconsistency, for example, journal names are sometimes abbreviated sometimes not.
The study is negligently written and there are many typing errors in the text. English grammar needs to be improved too.
Overall, the study contains valuable results that can be considered for possible publication after major revisions.
Some suggestions:
- L2: latin names should be in italic – in the title too.
- L21-24. Only general results are provided in the Abstract (only one sentence).
- L24: No conclusion was provided at the end of Abstract
- L130: Title of table 1 is not well informative. Give latin names in full.
- L177: Y axis should be in English. There is no X-axis title.
- L226: Sometimes you provide DOI sometimes not. And if you provide sometimes is black sometimes blue coloured.
The study is negligently written and there are many typing errors in the text. English grammar needs to be improved too.
Author Response
I send manuscript with corrections
Ernesto Cerna

Round 2
Reviewer 2 Report (New Reviewer)
Comments and Suggestions for Authors
Some progress were made, however, my major points were not changed (e.g. Intro, M and M, Dicussion, References).
In the Conclsuions, latin names should be in italics.
Comments on the Quality of English LanguageEnglish needs to be improved.
Author Response
Dear Reviewer,
This is a manuscript following the short communication article type, thank you.
This manuscript is a resubmission of an earlier submission. The following is a list of the peer review reports and author responses from that submission.
Round 1
Reviewer 1 Report
Comments and Suggestions for Authors
Dear Authors,
The present manuscript is a study that aim to provide alternatives to substitute the use of synthetic fungicides against Pythium amazonianum and Phytophthora cinnamomi. These two microorganisms, Pythium amazonianum and Phytophthora cinnamomi, are fungi that causes the Avocado tree wilt, and then issues for the Avocado production. The alternative suggested are extracts from Proboscidea parviflora W., and Phaseolus lunatus L. to control the development of mycelia of Pythium amazonianum. The idea looks very promising with couple of good extracts showing high in vitro inhibitory activity against the pathogen. In general, the manuscript is well written with very few mistakes that I have highlighted below. The discussion of the manuscript is incomplete. Answers for many questions needs to be addressed. For instance, what are the objectives on your research, why some extracts are better then other and why there are clear inhibitory differences of extracts against P. lunatus and P. parviflora? These discussion are the main point for the conclusion. However, essential experiments are missed. My concern about it is that to achieve this point on discussion the extracts needs to be qualitative and quantitative further characterised, for instance, by chemical analysis to determine total phenolics, total flavonoids and more. I hope it can be achieve soon.
Best regards,
Abstract -
Line 13 – Substitute “is aiming” to “aims” .
Introduction -
Line 31/32 – The sentence do note make sense. Please, review and rewrite this sentence to make it clear.
Line 33 – This sentence is not understandable. Please, review and rewrite this sentence to make it clear.
Line 44 – Please, substitute “created by” to “caused by”.
Line 51/55 – Please put all scientific names in italic. Seems that the reference numbers are substituting words in many sentences like in line 33, 51 and 56. Please, correct these mistakes along the text.
Line 56 – What is the importance of this ration between triterpenes and glucosiloxy-fatty acids from P. lousiana? Please, it should be clear. Explain it.
Methods -
How many replicates were used in the inhibitory assay experiment?
Line 81-98 – All this information is result. Please move it to item Results.
Line 88 – Please, substitute “63” by “Sixty-three” .
Line 98 - Please, substitute “100% of the raw” by “The raw”.
Line 101 – Please substitute “0.5cm of diameter explants” by “Explants with 0.5 cm of diameter”
Results -
Line 115 – Please, substitute “63” by “Sixty-three” .
Line 122 – Does “y” mean “and”? Please, correct it. Why did you not consider F.A.V.v.P.p. among the best extracts? At least it seems to be as satisfatory as F.A.T.P.l.
Line 130-131 – Why are you using ppm instead mg/L at this point? Please, standardize the measurements.
Line 147 – Please substitute “realized” by “observed”.
Line 167 - 173 – Why are 2200mg/L or 100mg/L used as a reference measure? It is not clear in the text, please explain that.
Reviewer 2 Report
Comments and Suggestions for Authors
The manuscript investigated the extracts from Proboscidea parviflora and Phaseolus lunatus on mycelial development of Pythium amazonianum. However, no chemical name or structure has been indentified from the extracts. So the manuscript may be too preliminary to be published.
Detailed comments:
1) All abbreviations should be introduced at the first time, including gene names, such as in Abstract. Line 135, what is "2E.F.P.p"? Whether is it different from "E.F.P.p"?
2) For all the tables and figures, no error bar or standard deviation was shown.
3) In Table 2 "E.F.P.p" extracts, 2000 mg/L showed a inhibition ratio of 4.43%; while 2104.75 mg/L showed a inhibition ratio of 92.60%. How was that possible?
4) Pay attention to significant digits, especially in Table 4.
Comments on the Quality of English LanguageThe language should be polished by an English native speaker. For example in Lines 18-19, I cannot understand "...showing that both extracts control and inhibit mycelial development, in particular P. lutatus..."
Round 2
Reviewer 2 Report
Comments and Suggestions for Authors
I cannot find a point-to-point response to my comments. No answer or revision has been made to the comments.
The manuscript investigated the extracts from Proboscidea parviflora and Phaseolus lunatus on mycelial development of Pythium amazonianum. However, no chemical name or structure has been indentified from the extracts. So the manuscript may be too preliminary to be published.
Detailed comments:
1) For all the tables and figures, no error bar or standard deviation was shown.
2) In Table 2 "E.F.P.p" extracts, 2000 mg/L showed a inhibition ratio of 4.43%; while 2104.75 mg/L showed a inhibition ratio of 92.60%. How was that possible?
3) Some standard deviations in Table 2 are zero! How was that possible?
4) Pay attention to significant digits, especially in Table 4. Too many numbers!
Comments on the Quality of English LanguageThe language should be polished by an English native speaker. For example in Lines 18-19, I cannot understand "...showing that both extracts control and inhibit mycelial development, in particular P. lutatus..."